# Biodegradation of Free Gossypol by *Helicoverpa armigera* Carboxylesterase Expressed in *Pichia pastoris*

**DOI:** 10.3390/toxins14120816

**Published:** 2022-11-22

**Authors:** Li Zhang, Xiaolong Yang, Rongzheng Huang, Cunxi Nie, Junli Niu, Cheng Chen, Wenju Zhang

**Affiliations:** College of Animal Science and Technology, Shihezi University, Shihezi 832000, China

**Keywords:** *cce001a*, *Helicoverpa armigera*, carboxylesterase, heterologous expression, gossypol detoxification

## Abstract

Gossypol is a polyphenolic toxic secondary metabolite derived from cotton. Free gossypol in cotton meal is remarkably harmful to animals. Furthermore, microbial degradation of gossypol produces metabolites that reduce feed quality. We adopted an enzymatic method to degrade free gossypol safely and effectively. We cloned the gene *cce001a* encoding carboxylesterase (CarE) into pPICZαA and transformed it into *Pichia pastoris* GS115. The target protein was successfully obtained, and CarE *CCE001a* could effectively degrade free gossypol with a degradation rate of 89%. When esterase was added, the exposed toxic groups of gossypol reacted with different amino acids and amines to form bound gossypol, generating substances with (M + H) *m*/*z* ratios of 560.15, 600.25, and 713.46. The molecular formula was C_27_H_28_O_13_, C_34_H_36_N_2_O_6_, and C_47_H_59_N_3_O_3_. The observed instability of the hydroxyl groups caused the substitution and shedding of the group, forming a substance with *m*/*z* of 488.26 and molecular formula C_31_H_36_O_5_. These properties render the CarE *CCE001a* a valid candidate for the detoxification of cotton meal. Furthermore, the findings help elucidate the degradation process of gossypol in vitro.

## 1. Introduction

Gossypol is a toxic phenolic compound that occurs naturally in cottonseeds [1]. Cottonseed meal is a major protein-rich byproduct of cotton processing and contains 38–65% crude protein [2]. It is considered an alternative protein source in animal feed due to its low cost and high content of protein, carbohydrates, and minerals [3,4,5]. The presence of gossypol restricts the use of cottonseed and its derivatives in animal feed [6]. The negative effects of gossypol on animal health have long been recognized. Furthermore, owing to the presence of a large number of microorganisms and soluble proteins in the rumen, free gossypol (FG) binds to proteins to form non-toxic bound gossypol, which has a remarkably more potent toxic effect on non-ruminants than on ruminants [7,8,9,10]. Animals may develop anorexia upon consumption of FG, which reduces productivity [11] and shows [12] hepatotoxicity [13] in the animals. This, in turn, leads to economic losses. The European Union Food Safety Authority (ESFA) recommends that the maximum concentration of FG in cottonseed meal administered as full-price feed be 1200 mg/kg. However, commercial cottonseed meal contains up to 7000 mg/kg of FG, which is far beyond the concentration range that exerts toxic effects animals can tolerate [14]. Therefore, the detoxification of FG is needed. Furthermore, these findings point to the necessity of eliminating FG in feeds.

Traditional methods have been used to reduce the toxicity of gossypol in cottonseed byproducts, including mechanical processing and chemical treatments, such as n-hexane extraction and Fe^2+^ methods [15]. Although the gossypol concentration is decreased by those methods, the quality of cottonseed protein, vitamin content, and feed palatability are also decreased [16]. Currently, the most popular method for detoxifying cottonseed meal is microbial solid fermentation by *Candida* sp. [17,18], *Aspergillus* sp. [19,20], *Rhizopus oryzae*, and *Mucor rouxii* [20]. Fungal metabolites, such as fungi and mycotoxins, produced during fermentation can be toxic to animals [21]. Hence, the safety of fermentation products used as animal feeds must be assessed [10]. The breakdown of FG in feed by enzymes is garnering significant interest [22]. Essentially, detoxification-related genes convert exogenous toxins through their encoded enzymes [23]. Enzymatic biocatalysts not only retain the ability to degrade gossypol but also do not introduce unwanted cells or their metabolites [24]. Therefore, the use of detoxifying enzymes with the potential to ensure feed safety and quality is a promising strategy. Surprisingly, to overcome the toxicity of gossypol, *Helicoverpa armigera* has developed a powerful detoxification system comprising detoxification enzymes such as CarE [25,26]. CarEs are α/β hydrolase proteins [27,28]. These are important detoxifying enzymes in insects and play an important role in the metabolism of toxic plant chemosensitive substances [29,30]. These enzymes have been associated with a series of reactions, demonstrate a wide range of substrate specificity, and play a role in the detoxification of exogenous substances, such as drugs and pesticides [31]. These detoxifying enzymes are primarily involved in the hydrolysis, sequestration, and binding of various plant chemosensitive substances [32]. An increase in CarE activity in *H. armigera* is observed with an increased dose of gossypol [33,34]. In contrast, when cotton leaves treated with high doses of the plant-protectant jasmonic acid (JA) are fed to *H. armigera*, the activity of CarE in *H. armigera* is decreased [35]. Transcriptomic data from Jin Minghui derived under conditions involving gossypol feeding showed that the expression of *H. armigera cce001a* (CarE gene) was significantly upregulated and silenced by RNAi technology, and the weight gain of *H. armigera* was significantly suppressed [36]. These findings suggest that CarEs play an important role in the metabolism of gossypol. However, in vitro degradation of gossypol by CarE has not been examined. We hypothesized that this gene may also be involved in the degradation of gossypol. 

This study explored the role of the *cce001a*-encoded CarE derived from *H. armigera* in gossypol detoxification. The *Pichia pastoris* expression system has the advantage of protein folding and post-translational modifications [37]. To this end, we established an efficient and safe expression system (Figure 1) to produce gossypol degrading enzyme (*CCE001a*), and then examined the ability of this protein to degrade gossypol and the putative intermediate metabolic pathway for gossypol degradation, thereby reducing the risk of toxic effects of gossypol on animals.

## 2. Results and Discussion

### 2.1. Source of CarE and Construction of Expression Vector

The amino acid sequence of the protein of CarE *CCE001a* (GenBank accession number: HM191471.1) isolated from the intestine of *H. armigera* was selected. Bioinformatics analysis showed that the gene comprised an open reading frame (ORF) of 1668 nucleotides encoding 555 amino acid residues. The gene product showed a molecular weight of 62.81 kDa, an isoelectric point of 5.32, and an instability coefficient of 35.22. Based on these findings, it was considered a stable protein. The average value of hydrophilicity (GRAVY) was found to be −0.262, indicating an overall hydrophilic protein. The presence of a signal peptide at amino acid position 17 indicated that the protein may be secreted (Appendix A). Signal peptide sequences are important for the identification and characterization of new candidates [38], which serve to secrete proteins. These proteins are then guided by signal peptides across the cell where they are synthesized in other tissue cells [39]. The target protein is more readily secreted and expressed in the supernatant in the presence of the signal peptide, which would reduce the time consumption [40,41]. PCR products were amplified and cloned into the *EcoRI* and *XbaI* sites of pPICZαA (Appendix A), which was verified by double digestion, and two bands were visible with bands of 1668 bp and 3500 bp (Figure 2), and a recombinant plasmid pPICZαA-*cce001a* was constructed. Sequencing results verified the presence and correct orientation of the *cce001a* ORF, laying the foundation for subsequent expression.

### 2.2. Expression of Recombinant CarE CCE001a

The most convenient method to characterize the ability of CarEs to degrade FG involves the use of recombinant enzymes [42]. *H. armigera cce001g* [43], *cce001d*, *cce001h*, *cce016a*, and *cce001j* [44] have been expressed using prokaryotic expression and Baculovirus systems. However, since the prokaryotic system expression is prone to protein folding errors [45], *cce001a* was not expressed successfully in previous studies. 

The recombinant yeast genome was extracted as a template using 5′ *AOX* and 3′ *AOX P. pastoris*-specific primers (Figure 3a) to obtain a 1668 bp band. A 500 bp band was visible when a segment of the exogenous gene was used as a primer (Figure 3b). The exogenous gene was successfully integrated into the *P. pastoris* genome. To confirm the expression and secretion of recombinant enzymes, the expression supernatant was examined by SDS-PAGE after methanol stimulation, and a distinct band with a molecular weight of 76 kDa was observed at 72 h and 96 h (Figure 4a). No target protein was detected in the empty pPICZαA-GS115 plasmid. The expressed target protein was larger in size than the predicted value of the target protein due to the presence of one potential n-glycosylation site (NetNGlyc) [46,47] in *CCE001a* (Appendix A). Western blotting revealed a specific His-tag band corresponding to *CCE001a* (Figure 4b), whereas no His-tag band was detected in the *GS115* strain transformed with empty pPICZαA vector. This finding confirmed that the target protein was successfully expressed in *P. pastoris*.

### 2.3. Exploration of CCE001a’s Activity on Model Substrates

To evaluate the functional characteristics of the recombinant protein, it was necessary to validate the activity. The effects of CarEs, such as *H. armigera* CarE expressed in *E. coli* [48,49,50,51] and 14 types of *H. armigera* CarEs obtained from baculovirus sf9 on the model substrate 1-naphthol, have been evaluated [44]. We also evaluated the enzyme activity of the recombinant *CCE001a* protein with the model substrate alpha-naphthyl acetate, and the protein concentration was determined using Bradford’s method [52]. At a wavelength of 450 nm, the absorbance of CarE increased with time within a certain time range. The absorbance was the highest at 150 s (Figure 5), The enzyme activity of the recombinant protein was shown to be 145.05 nmol/min/mg.prot of protein, which was measured at 450 nm.

### 2.4. Gossypol Analysis

The degradation rate of FG is the most prominent indicator of changes in gossypol content. During the whole reaction, it is most important to detect the change in FG, and the total amount of gossypol is represented by the sum of free and bound gossypol [53]. Many studies tend to ignore the changes in total gossypol (TG) levels. In this study, in order to rule out the possibility that changes in gossypol levels can be attributed to physical action rather than biodegradation, we detected TG levels in the system. The traditional aniline method for the determination of gossypol was not used due to the low recovery of gossypol and a complex peak formed by the combination of gossypol and N, N-dimethylformamide (DMF) [54]. FG remaining in the reaction mixture was measured using the HPLC method with the aforementioned modifications [55,56]. In this study, the content of FG and TG were determined systematically. Since gossypol is unstable and easily oxidized, the addition of NADPH-Na4 initiates the reaction and stabilizes gossypol [55]. The results are shown in Table 1. The results of the blank group showed that the levels of total and FG decreased from 450 µg to 406.62 µg due to spontaneous oxidation over time even without the action of the enzyme (Table 1). This finding can be attributed to the inherent instability of gossypol. Although no plasmid was inserted in the control group, *P. pastoris* GS115 itself expressed certain endogenous enzymes. Hence, TG and FG levels decreased, which was consistent with the results of previous studies [57]. Under the influence of recombinant CarE, TG and FG levels decreased rapidly in the experimental group. The degradation rate of FG was 89%, and the difference between the experimental group and the control group were extremely significant (*p <* 0.01). Previously, a *Lactobacillus* strain screened from the rumen of dairy cows demonstrated the best degradation effect on gossypol. With an increase in the time of microbial action on gossypol, the degradation rate reaches 83% [9]. The enzymatic degradation effect of exogenous substances is better than that of microorganisms, and the time is shorter. Overall, finding an efficient and safe enzyme provides a theoretical basis for the degradation of gossypol. The biodegradation effect of recombinant CarE on gossypol was evaluated. Furthermore, FG was converted into degradation products, which laid the foundation for the subsequent LC-MS identification of its properties.

### 2.5. Mass Spectrometry Analysis of Intermediate Products

CarE can degrade gossypol, Hence, an evaluation of its structure is necessary. Degrading gossypol is a process which aims to mitigate the toxicity of FG by converting it into less toxic or non-toxic substances. One way to reduce the toxicity of gossypol and enhance its biological properties is to convert it into azepine derivatives, such as Schiff bases or hydrazones. The Schiff base or hydrazone gossypol can undergo Schiff base formation [58,59], ozonation [15], oxidation [60], and methylation [61] to form gossypol derivatives.

CarEs is a serine hydrolase that reacts with many compounds with different structures that contain ester bonds [62,63,64,65,66]. As mentioned above, the toxicity of gossypol can be attributed to six phenolic hydroxyl groups and two aldehyde groups, and CarEs can catalyze the hydrolysis of esters or amide compounds into corresponding alcohols and carboxylic acids [67]. CarEs are widely considered attractive and advantageous biocatalysts due to their characteristics, such as their ability to accept a wide range of substrates and their high stereospecificity, high tolerance to organic solvents, and lack of cofactors required for the reaction [68]. Conserved region prediction analysis was performed with the *H. armigera* CarE *CCE001a* protein (https://www.genome.jp/tools/motif//, 18 April 2022). The protein showed a co-esterase family (PF00135) domain and a dehydrogenase family (PF07859) domain (Appendix A). The enzymatic reaction itself is a complex process involving graded metabolites. Hence, we initially hypothesized that the action of CarEs on gossypol may involve the hydrolase activity of the enzyme and the reaction of unknown intermediate metabolites. Gossypol is metabolized in different pathways in animals such as pigs [69] and hens [70]; however, these metabolites have not been completely elucidated as gossypol and its derivatives are excreted in low concentrations in animal feces [71,72]. Gossypol is bioconverted to gossypol ketone, gossypol acid, and demethylated gossypol acid [71]. The main scavenging mechanisms of gossypol include glucoaldehyde acidification and bile excretion [72]. In our reaction, CarE showed the characteristics of a hydrolase and some unknown enzymatic functions. The phenolic hydroxyl and aldehyde groups of gossypol were exposed, and the stability was relatively poor. Therefore, we explored the possibility of another degradation mechanism of gossypol. Metabolites were identified from the reaction system using UPLC-QTOF/MS and Masslynx 4.1 software (Waters Corporation, Milford, MA, USA) analysis and divided into standard, control, and test group. Gossypol formed a deprotonated molecule [M-H]^−^ at *m*/*z* 518.1857 with a theoretical mass of 517.1910 in the negative ion scan mode. A comparison of the total ion current of the three groups (Figure 6) shows that the peak observed in the test group (which included the recombinant CarE in the reaction) at a retention interval of 19 min corresponds to gossypol, and its lower signal intensity quantitatively indicates that the gossypol content decreased under the influence of the recombinant CarE. In the endogenous enzyme group, the gossypol level was lower than that of the standard product. Although no recombinant CarE was added, the endogenous enzymes of *P. pastoris* may bind with gossypol to form bound gossypol, which reduced the TG content. After recombinant CarE was added to the test group (Figure 7), a new peak was observed that was not seen in the results from the standard and control groups. The peak had a retention time of 0.75 min and was measured at *m*/*z* 268.53. The chemical formula was C_19_H_24_O. We designated it as compound M0. The theoretical mass was 269.19, and the product was confirmed to be hemigossypol [73]. Gossypol itself is a polyphenol binaphthyl, which endows the binaphthyl bond with instability due to photosensitivity [74]. Another peak in the test group was observed at a retention interval of 4.51 min with *m*/*z* as 487.01 This compound, designated M1, had a chemical formula of C_31_H_36_O_5_ and a theoretical mass of 489.26. It was also the product of recombinant CarE activity, during which the exposed groups of gossypol were removed to detoxify gossypol. The degradation of gossypol by enzymes is less frequently examined. Previous studies have focused on the oxidation of gossypol by *H. armigera* P450 enzymes [57]. Laccase cyclization-hydroxyl aldehyde condensation [61] is a reaction of gossypol that renders it less toxic. This finding also coincides with the fact that enzymes have a wide range of substrates. Another detoxification product, M2, appeared at a retention time of 4.97 min, *m*/*z* of 600.03. This substance, with a chemical formula of C_34_H_36_N_2_O_8_ and theoretical mass of 601.25, has not been observed in other studies. Compound M2 was formed owing to the effect of CarE on the toxic aldehyde group of gossypol. This reaction formed an intermediate carboxylic acid in place of the aldehyde. Further, hydrolysis of the carboxylic acid derivative realized the complete elimination of the aldehyde group. Another detoxification product corresponding to the peak at 5.28 min was designated M3 with an *m*/*z* of 713.05. The structural formula of M2 was C_47_H_59_N_3_O_3_, with a theoretical mass of 714.46, and it is considered to be an Aza derivative of gossypol. We analyzed the recombinant CarE and found that the alanine and leucine levels were the highest. Furthermore, the aldehyde group on gossypol could react with the α-NH_2_ of amino acids to form Schiff bases, thereby forming azides. This finding is consistent with those of previous reports, which showed that binding or removing toxic aldehyde groups from gossypol can effectively reduce its toxicity. The metabolic pathway for the in vitro degradation of gossypol is shown in Figure 8. The exact mass, elemental composition, and molecular formula of the gossypol degradation products are listed in Table 2. In general, the metabolism of gossypol by the enzymatic reaction of *H. armigera CCE001a* is a complex process involving hydrolysis, dehydrogenation, and covalent binding to amine products. Hence, defining the metabolism of gossypol at all levels is challenging.

## 3. Conclusions

In this study, a CarE from *H. armigera*, *CCE001a*, was successfully expressed in *P. pastoris* GS115, and the activity of recombinant CarE on FG was examined. After treatment with recombinant CarE, the degradation of TG was 90%, and the degradation of FG was up to 89%. Detoxification is realized via radical and hydroxyl group attacks or reduction in gossypol levels mediated via binding with amino acids of the CarE to form azide compounds. This study confirms that recombinant CarE isolated from *H. armigera* can be utilized as an effective gossypol-degrading enzyme for cottonseed meal, a high-protein animal raw material. This enzyme would aid in detoxifying the cottonseed meal, allowing it to be used as animal feed.

## 4. Materials and Methods

### 4.1. Source of Target Gene Sequence

*H. armigera* was collected from cotton-growing sites in Shihezi, Xinjiang Autonomous Region, China. *H. armigera* was reared on a gossypol diet under laboratory conditions. Total RNA was extracted from the gossypol-treated fifth-generation larvae of *H. armigera* using TRIzol reagent (Sigma-Aldrich, St. Louis, MO, USA). The extracted RNA was transcribed into cDNA following the manufacturer’s protocol (Promega, Madison, WI, USA).

### 4.2. cce001a Cloning and Expression Vector Construction

The *cce001a* gene (GenBank accession number: HM191471.1) was amplified, using TaKaRa Ex Taq^®^ (TaKaRa, Dalian, China) with degenerate primers *001a* F (ATGTCAGACAGCGCACAGGACG) and *001a* R (ACTCCATACATCTGCTGAATAT). Restriction sites were introduced using cDNA extracted from the midgut of *H. armigera* as the template. Polymerase chain reaction (PCR) cycling conditions are listed as follows: one cycle at 94 °C for 3 min, followed by 35 cycles at 94 °C for 30 s, 60 °C for 30 s, and then 72 °C for 2 min. The PCR product was purified using an EZNA Gel Extraction Kit (Omega BioTek, Norcross, GA, USA), cloned into the pGEM-T vector, and transformed into *E. coli* Stbl3-competent cells for two-way sequencing (Shanghai Jierui, Shanghai, China). After the expression vector was digested with EcoRI and XbaI, the PCR product and expression vector were ligated using a Gibson assembly. The plasmid was transformed into *E. coli* DH5α-competent cells, extracted, and sent to Shanghai Jierui for sequencing.

### 4.3. Electrotransformation, Screening, and Identification of Transformants with High Expression of CCE001a in P. pastoris

A total of 5 μg of the pPICZαA-*CCE001a* plasmid was linearized by the SacI rapid digestion method and electroporated into *P. pastoris* GS115-competent cells (80 μL) under the following conditions: 25 μF, 200 Ω, and 1.5 kV using a micropulser electroporator (Bio-Rad, Hercules, CA, USA). Following this, 1 mL of pre-cooled sorbitol (1 M) was added to the electro-rotor cup and the cells were incubated at 28 °C for 2 h. The cells were then spread on YPDS (Yeast Extract Peptone Dextrose Medium Sorbitol)-Zeo(Bleomycin) plates containing 100 mg/mL Zeo and cultured at 28 °C for 48–72 h. To screen for high-copy recombinant clones, the cells were incubated in the presence of 0.5–3.0 mg/mL Zeocin^®^ at 28 °C until colonies were visible. Resistant clones were selected and the yeast genome was extracted. Using the genome as a template and the universal primers *5′AOX* (GACTGGTTCCAATTGACAAGC) and *3′AOX* (GCAAATGGCATTCTGACATCC), positive recombinants were screened by PCR. Positive clones were selected and inoculated into a medium containing 100 mM potassium phosphate (pH 6.0), 0.34% yeast nitrogen base, 0.00004% biotin, 1% yeast extract, 2% peptone, and 1% glycerol. The mixture was incubated at 28 °C with shaking (250 rpm) for 24 h. Cell pellets were collected, resuspended in buffered methanol-complex medium (BMMY), and incubated again at 28 °C with shaking (250 rpm), with the addition of methanol every 24 h to maintain its content at 1% of the total volume. The total time required for continuous fermentation in the shake flask was 120 h. Samples were derived every 12 h and stored at −80 °C for subsequent use.

### 4.4. Sodium Dodecyl Sulfate-Polyacrylamide Gel Electrophoresis (SDS-PAGE) and Western Blotting

Cells stimulated with methanol were collected at various time points during their incubation, concentrated by centrifugation at 12,000× *g* at 4 °C, and then disrupted by ultrasonication (Thermo Fisher Scientific, Waltham, MA, USA). Proteins in the supernatant were precipitated with trichloroacetic acid, boiled for 10 min and mixed with an equal volume of 1× loading buffer. SDS-PAGE was performed on a 10% polyacrylamide gel and Coomassie Brilliant Blue R250 was used to detect the protein bands. After transfer onto a polyvinylidene fluoride membrane, CarE was detected with an anti-His mouse polyclonal antibody (Cell Signalling Technology, Danvers, UT, USA) and diaminobenzidine substrate (Shenggong, Shanghai, China).

### 4.5. CarE Activity Measurements

CarE activity was determined by measuring the conversion of α-naphthyl acetate to 1-naphthyl ester. Assay reagents were mixed as specified in Table 1 in a 96-well plate and CarE activity was calculated by measuring the absorbance at 450 nm. Two reactions were set up, namely, one containing the enzyme solution and one blank with distilled water. Absorbance (A1) was measured every 10 s after mixing the reagents for 5 min at 37 °C, and absorbance (A2) was measured after 310 s. ΔA was calculated by measuring the difference between A1 and A2. The enzyme activity unit was defined as an increase of 1 unit in the catalytic light absorption value per min per mg of protein and per mL of the reaction system at 37 °C. The following formula was used to determine the U/mg:CarE enzyme activity (U/mg protein) = 8 × (△A measuring tube − △A blank tube) × Cpr × F.

### 4.6. Ultra-High Performance Liquid Chromatography (UHPLC) Analysis of Gossypol

Active recombinant CarE was expressed in *P. pastoris* GS115 and UHPLC was used to analyze free gossypol after treatment with this enzyme. Free gossypol (20 μL at 300 μg/mL) was added to 200 μL of phosphate-buffered saline (PBS). Approximately 100 μg of protein was added to the mixture, which was then transferred to 1.5 mL tubes. To start the reaction, 4 μL NADPH-Na_4_ (25 mM) was added, and the mixture was incubated in the dark at 37 °C for 1.5 h. To stop the reaction, 400 μL of 70% acetone aqueous solution was added. An ultrasonic cleaner was used for sample extraction over a period of one hour with intermittent turning to maximize the extraction. Samples were centrifuged at 12,000× *g* for 1 min at 4 °C and the supernatant was collected. The supernatant was analyzed by HPLC [42], with total gossypol determined as described previously. The following groups were analyzed: blank group (BL-1), control group (Co-1), test group (MY-1).

The gossypol standard stock solution was diluted with 70% acetone aqueous solution to prepare gradient concentrations of 1–50 μg/mL. A total of 4 μL NADPH-Na_4_ was added to each reaction to evaluate the best chromatographic conditions and compare the peak area Y with concentration X. Linear regression was performed to obtain the standard reaction equation. A SUPELCOSIL^TM^ C18 column (4.6 mm × 250 mm, 5 μm; Sigma-Aldrich) and a 20-μL loop were used for chromatographic separation at 25 °C. Acetonitrile and 0.3% formic acid were used as mobile phases A and B, respectively. The gradient was linearly increased to 75% A and 25% B within 10 min, maintained at 85% A and 15% B for 10 min, and finally returned to 60% A and 40% B. The injection volume was 20 μL, the column temperature was maintained at 25 °C, and the detection wavelength was set at 380 nm. No less than three replicates per sample were evaluated.

### 4.7. UPLC-Quadrupole Time of Flight Mass Spectrometry (UPLC-QTOF/MS) Analysis

UPLC-QTOF/MS was used to determine whether recombinant CarE degraded gossypol and its resulting metabolites. The samples subjected to treatments included the control group (no enzymes added), endogenous enzyme group, and test group (esterase protein). The reaction mixture was incubated with gossypol in PBS (pH 7.0) at 30 °C for 30 min in a water bath. Next, 450 μL of methanol (−20 °C) was added, and the solution was vortexed for 30 s and centrifuged at 12,000 rpm and 4 °C for 10 min. The supernatant was transferred to a new 1.5 mL centrifuge tube, to which 300 μL acetonitrile was added. The tube was centrifuged at 12,000 rpm and 4 °C for 10 min, and 200 μL of the supernatant was passed through a 0.22 μm filter. The resulting sample was subjected to UPLC-QTOF/MS.

#### 4.7.1. UPLC Conditions

An ACQUITY UPLC HSS T3 column (100 mm × 2.1 mm, 1.8 μm; Waters, Milford, MA, USA) was used with a flow rate of 0.3 mL/min, injection volume of 5 μL, and column temperature of 35 °C. Methanol (chromatographic grade) and acetonitrile (chromatographic grade) solutions were used as mobile phases A and B, respectively. Gradient elution was maintained as follows: 0 to 2.00 min with 20% B, 2.00 to 5.00 min with 40% B, 5.00 to 8.0 min with 60% B, 8.00 to 12 min with 80% B, and 12.00 to 22.00 min with 100% B.

#### 4.7.2. MS Conditions

An electrospray negative ion mode was used as the ion source and the ion transfer tube temperature was 320 °C. The quantitative detection method was set to full-scan mode. The resolution was set to full scan (full mass) and secondary mass spectrometry scanning (MS/MS) was performed at 17,500. The isolation window (n window) was 1.0 *m*/*z*, the electrospray voltage was set to 3500 V, and the sheath gas pressure was set to 275.8 kPa. The auxiliary gas rate, backflush gas pressure, auxiliary gas heating temperature, and RF prism voltage were 180 L/h, 13.8 kPa, 300 °C, and 50%, respectively.

### 4.8. Data Analysis

One-way ANOVA was performed using SPSS 11.5 (SPSS Inc., Chicago, IL, USA), and Duncan’s method was used for significant difference analysis (*p* < 0.05). All data were expressed as mean  ±  SEMs. Mass spectral data were analyzed by Masslynx 4.1 (Waters) based on full scan analysis and extracted ion chromatograms. ChemDraw software (CambridgeSoft, Cambridge, MA, USA) was used to draw chemical structural formulas.

## Figures and Tables

**Figure 1 toxins-14-00816-f001:**
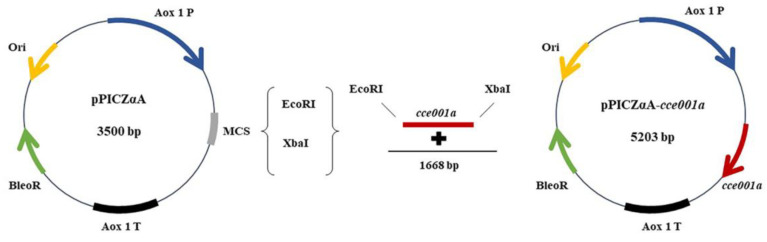
Plasmid maps of pPICZαA and pPICZαA-*cce001a*. Aox1P blue arrow promoter is the direction of methanol-induced protein expression; Ori yellow arrow refers to the replication initiation site; gray box is the polyclonal site; black arrow is the transcription terminator; green arrow is the antibiotic marker; red arrow is the *cce001a* gene. Cloning of the *cce001a* fragment into the polyclonal site (MCS) of the linearized expression vector pPICZαA by Gibson assembly. Assembly with pPICZαA to form a complete construct.

**Figure 2 toxins-14-00816-f002:**
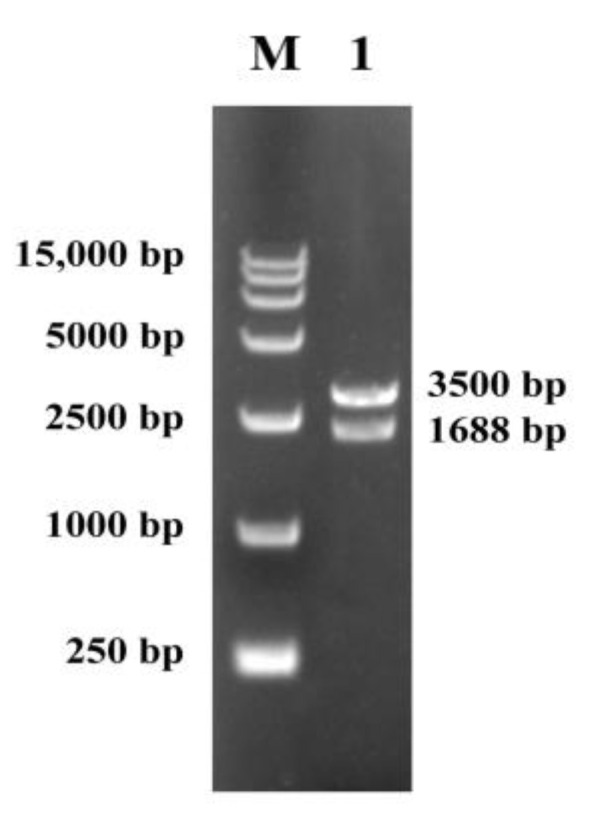
DNA marker (lane M); pPICZαA-*cce001a* verified by *EcoRI* and *XbaI* double digestion (lane 1).

**Figure 3 toxins-14-00816-f003:**
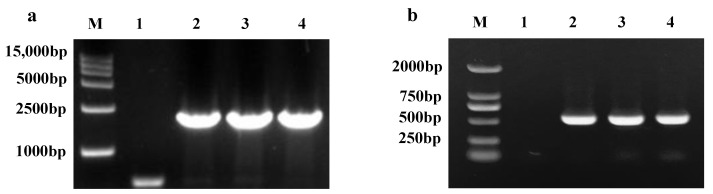
(**a**) DNA marker (lane M),Empty pPICZαA (lane 1), pPICZαA-*cce001a* (lane 2, 3, 4). The PCR templates of empty pPICZαA and pPICZαA-*cce001a* involved recombinant GS115 genome fragments. The following primers were used: 5′AOX and 3′AOX primers (lanes 2, 3, and 4). (**b**) Empty pPICZαA (lane 1), pPICZαA-*cce001a* (lane 2, 3, and 4). PCR templates of empty pPICZαA plasmid and pPICZαA-*cce001a* comprised recombinant GS115 genome fragments, and *cce001a* specific primers were used (lanes 2, 3, and 4).

**Figure 4 toxins-14-00816-f004:**
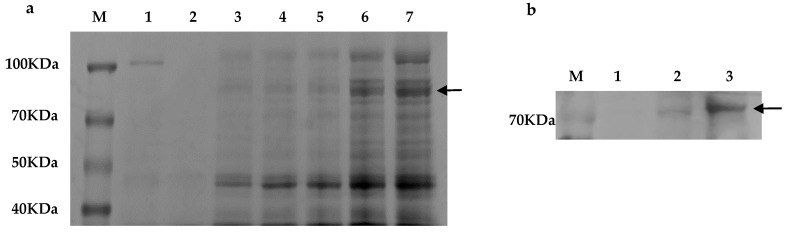
Secretion-induced protein expression and Western blot analysis of pPICZαA-*CCE001a* recombinant strain. (**a**) The sample was separated on a 10% SDS-PAGE gel; lane M represents the protein molecular marker, Lane 1 specifies the pPICZαA-GS115 control, lanes 2–7 correspond to the expression supernatant of GS115-pPICZαA-*CCE001a* in the time range of 12 h to 96 h; lane 2 = 12 h, lane 3 = 24 h, lane 4 = 36 h, lane 5 = 48 h, lane 6 = 72 h, and lane 7 = 96 h. (**b**) Western blot analysis of recombinant CarE using anti-His tag antibody. M represents the protein molecular marker, lane 1 is the Western blot analysis of control, lane 2 and lane 3 are the Western blot analysis of recombinant CarE.

**Figure 5 toxins-14-00816-f005:**
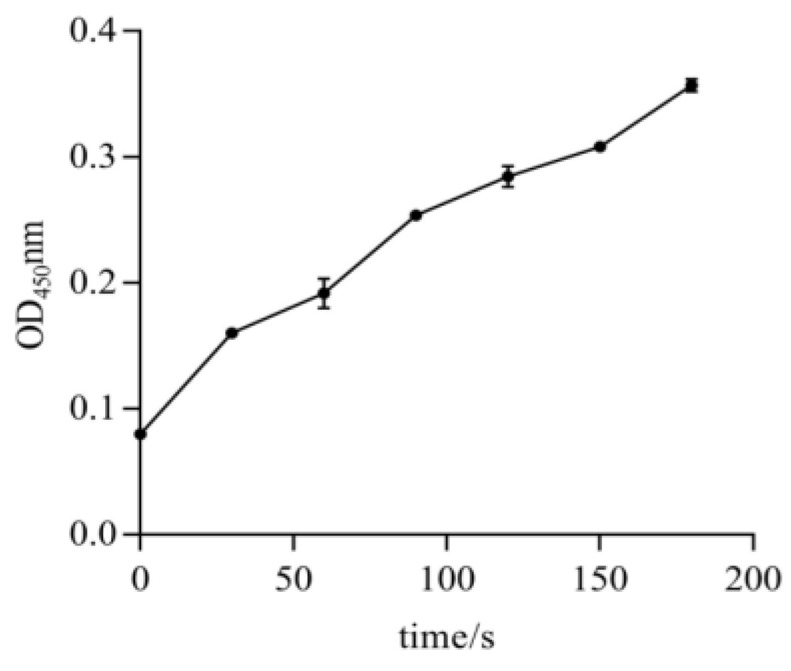
Absorbance of CarE at OD_450_ nm over time.

**Figure 6 toxins-14-00816-f006:**
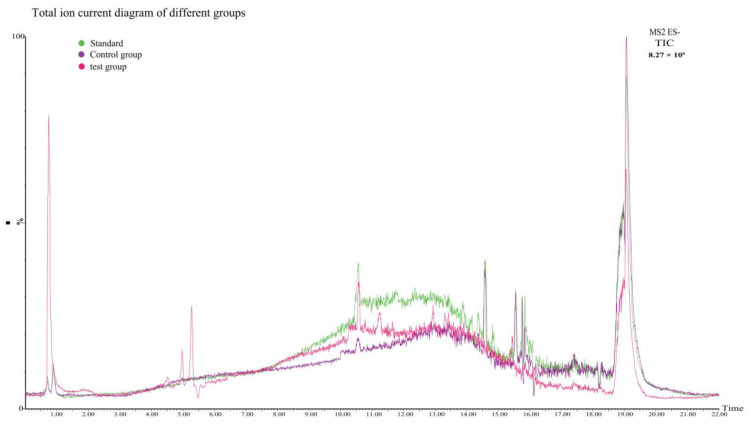
Total ion current diagram demonstrating various experimental groups. Green represents the gossypol standard, purple represents the control group, and red represents the test group.

**Figure 7 toxins-14-00816-f007:**
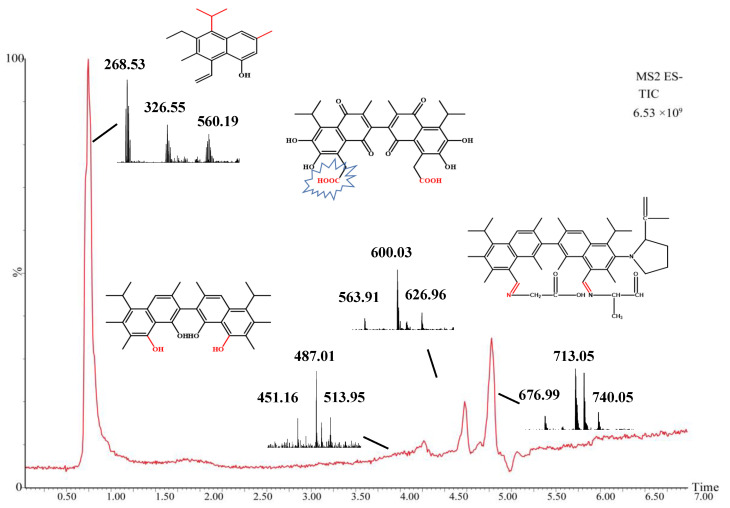
TIC and mass spectrum of gossypol metabolites detected by UPLC-QTOF/MS under negative ion scanning. The total ion current diagrams of the three groups were compared, and the substances appearing in the test group, but not in the other groups, were analyzed further. Due to the instability of gossypol, it was degraded to semi-gossypol and its derivatives. GS115-pPICZαA-*CCE001a* esterase group used gossypol as a substrate. Furthermore, nitrogenous substances in degrading enzymes can interact with gossypol and metabolize it. The products of interest formed in the reaction showed *m*/*z* ratios of 268.18, 600.25, 713.46, 560.15, and 488.26.

**Figure 8 toxins-14-00816-f008:**
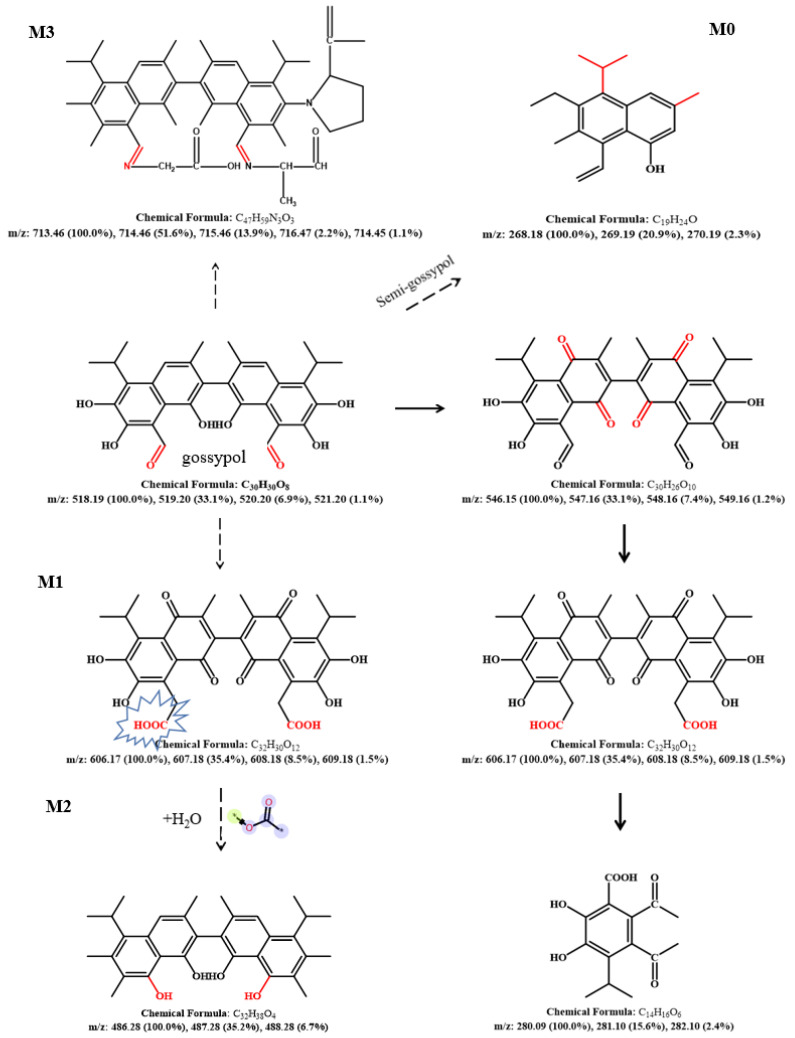
Schematic diagram of the degradation pathway of gossypol mediated by the *CCE001a* enzyme of *H. armigera.* Note: The dotted arrow indicates the intermediate product of gossypol obtained after the action of esterase and gossypol, “*” indicates different types of functional groups and the solid arrow indicates the metabolic pathway of gossypol [75] reported in a previous study.

**Table 1 toxins-14-00816-t001:** Measurement of gossypol levels and degradation rate after the addition of recombinant CarE.

Grouping	Gossypol Content (μg/mL)		Detoxification Rate (%)	
	TG	FG	TG	FG
Bl-1	406.62 ± 0.09 ^a^	258.63 ± 0.01 ^a^	10	42
Co-1	85.76 ± 0.07 ^b^	48.15 ± 0.07 ^b^	80	81
MY-1	42.34 ± 0.05 ^c^	27.43 ± 0.03 ^c^	90	89

Note: In Table 1 BL-1, Co-1, and MY-1 represent the blank group, control group, and test group, respectively. Values are presented as mean ± SEMs. Different lowercase letters in the same column indicate significant differences (*p* < 0.05). TG, total gossypol; FG, free gossypol.

**Table 2 toxins-14-00816-t002:** The mass spectra data for gossypol and its intermediates.

Compound	Experimental Mass (*m*/*z*)	Theoretical Mass (*m*/*z*)	Retention Time (min)	Molecular Formula
Gossypol	517.18	518.19	19.7	C_30_H_30_O_8_
M0	268.18	269.19	0.75	C_19_H_24_O
M1	488.22	489.26	4.97	C_31_H_36_O_5_
M2	600.25	601.25	4.97	C_32_H_30_O_12_
M3	713.46	714.46	5.28	C_47_H_59_N_3_O_3_

## Data Availability

The data that support the findings of this study are available from the corresponding author upon reasonable request.

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
