# Peer review of "Biodegradation of Free Gossypol by Helicoverpa armigera Carboxylesterase Expressed in Pichia pastoris"

_toxins, 2022, doi:10.3390/toxins14120816_

Round 1

Reviewer 1 Report

The manuscript describes the degradation of gossypol, a toxic compound present on cottonseeds, through a recombinant carboximetilesterase from Helicoverpa armigera. The work presented here focuses on a very interesting objective, and it is clear that a lot of experimental work has been performed by the authors. I suggest the following changes to improve the quality of the manuscript

Title: Helicoverpa armigera and Pichia pastoris should be written in italics. Please, check the rest of the manuscript (For example, line 53, 103, 104, 109 etc)

Introduction: Please, correct punctuation (Ex: line 25, 28, 55, 72, etc) and grammar errors (Ex: 72, 80, etc). Genes should be written in italics.(Ex 86., 96).

001G, 104 001f, 016a and 001d refers to genes? (line 105)

Results:

-For better comprehension, I recommend writing results and discussion separately

-Lines 109-111 are repeated on results, in lines124-126, but with different references, please rewrite.

-Line 103 says: “It is known that 14 H. armigera carboxylesterases have been characterized in the E. coli expression system, such as 001G, 104 001f and 016a[28] and 001d[25], but 001a has not been expressed, whereas on line 140 says “Genes of H. armigera carboxylesterase have been successfully expressed, including 140 CCE001d, CarE001G, CCE001a, and CCE001H[36], using an E. coli expression system” Please, check and correct.

-Line 194. Please, explain which means free and bound gossypol

_Line 222: Please, rewrite: Table 2: Determination of free gossypol (FG) and total gossypol (TG) in different groups.

Reviewer 2 Report

It is important to reduce the toxicity of gossypol from cotton seed by-products. The present study provided a possible candidate, CCE001a, for degradation of gossypol in vitro. It is a meaningful study but needs substantial modification before publication.

1.        Line 78-80: Since six genes were upregulated, why was only CCE001a selected for further study but not the other five genes?

2.        Section 2.1: The paragraph introduces the source of carboxylesterase and construction of expression vector, it seems more like a background introduction but not RESULTS and DISCUSSION. Some descriptions are repetitive to the introduction section. It is suggested to discuss with the results of the present study.

3.        Line 124-131: This paragraph is also repetitive description. P450 enzymes play the most important role in the degradation of gossypol, why choose CarE instead of P450?

4.        Line 132-147: Again, repetitive introduction is here. It is suggested to provide the results of present study and then to discuss.

5.        Fig. 3a: which is the target protein in the gel? Fig. 3b: Needs to provide control for western blot analysis.

6.        Writing needs a lot of improvement, especially INTROCUTION, RESULTS AND DISCUSSION sections.

Round 2

Reviewer 2 Report

1.       The authors did not explain clearly why CCE001a was chosen as the target of the study, and the INTRODUCTION describes the relationship between detoxification and resistance, which is not relevant to the study. It is suggested that the authors make a major revision to the INTRODUCTION to make clear the background and necessity of the study

2.       The key word for “helicoverpa armigera” should be “Helicoverpa armigera”. The first letter of some keywords is uppercase, some is lowercase, need to be consistent.

3.       Where is the data for 2.1? where is the description for Figure 1? Too much irrelevant content was described in section 2.2, it is suggested to give a major revision.

4.       Fig. 3b: which is the target protein in WB gel? The electrophoretic band in Fig. 3b is not clear, it is suggested to replace it with another clearer picture.

5.        Figure 4 is not clear and error bars in Figure 4 are not clear. Needs to be revised.

6.       I don’t think Table 1 is necessary.

7.       More details should be added to 3.10 Data analysis.

The language needs to be modified by native English speakers.

Round 3

Reviewer 2 Report

1.       Line 32-33: “…and shows and hepatotoxicity in the animals…”, delete “and” after “shows”.

2.       Line 66-68: The genus name in the second occurrence of the scientific name should be abbreviated. There were a number of errors in the manuscript.

3.       Line 73-83: This paragraph needs to be condensed.

4.       S1: The headline for S1 is improper. Whether S1 is a figure or a table?

5.       I didn’t find a reference to Figure in the text.

6.       There are also quite a few formatting and writing problems.
